

# Intra-colonial diversity in the scleractinian coral, *Acropora millepora*: identifying the nutritional gradients underlying physiological integration and compartmentalised functioning

Jessica A. Conlan[1], Craig A. Humphrey[2], Andrea Severati[2] and David S. Francis[1]

[1] School of Life and Environmental Sciences, Deakin University, Warrnambool, Victoria, Australia
[2] The National Sea Simulator, Australian Institute of Marine Science, Townsville, Queensland, Australia

## ABSTRACT

Scleractinian corals are colonial organisms comprising multiple physiologically integrated polyps and branches. Colonialism in corals is highly beneficial, and allows a single colony to undergo several life processes at once through physiological integration and compartmentalised functioning. Elucidating differences in the biochemical composition of intra-colonial branch positions will provide valuable insight into the nutritional reserves underlying different regions in individual coral colonies. This will also ascertain prudent harvesting strategies of wild donor-colonies to generate coral stock with high survival and vigour prospects for reef-rehabilitation efforts and captive husbandry. This study examined the effects of colony branch position on the nutritional profile of two different colony sizes of the common scleractinian, *Acropora millepora*. For smaller colonies, branches were sampled at three locations: the colony centre (S-centre), 50% of the longitudinal radius length (LRL) (S-50), and the colony edge (S-edge). For larger colonies, four locations were sampled: the colony centre (L-centre), 33.3% of the LRL (L-33), 66.6% of the LRL (L-66), and the edge (L-edge). Results demonstrate significant branch position effects, with the edge regions containing higher protein, likely due to increased tissue synthesis and calcification. Meanwhile, storage lipid and total fatty acid concentrations were lower at the edges, possibly reflecting catabolism of high-energy nutrients to support proliferating cells. Results also showed a significant effect of colony size in the two classes examined. While the major protein and structural lipid sink was exhibited at the edge for both sizes, the major sink for high-energy lipids and fatty acids appeared to be the L-66 position of the larger colonies and the S-centre and S-50 positions for the smaller colonies. These results confirm that the scleractinian coral colony is not nutritionally homogeneous, and while different regions of the coral colony are functionally specialised, so too are their nutritional profiles geared toward meeting specific energetic demands.

Corresponding author
Jessica A. Conlan,
conlan@deakin.edu.au,
jessconlan@live.com.au

## INTRODUCTION

Scleractinian corals are colonial organisms comprising multiple asexually produced and physiologically integrated polyps (*Oren et al., 2001*; *Bone & Keough, 2010*). The colonial nature of scleractinian corals is highly beneficial, with modular coral colonies being able to attain large increases in size and volume isometrically, allowing the component units to remain small (*Vollmer & Edmunds, 2000*; *Hughes, 2005*). This reduces the chance of whole colony mortality and increases colony exposure to exploitable environmental factors, such as sunlight and nutrition sources (*Nakamura & Yamasaki, 2006*; *Bone & Keough, 2010*).

Physiological integration also permits resource translocation from branches growing in favourable microhabitats to those growing under more adverse conditions (*Hemond, Kaluziak & Vollmer, 2014*; *Nozawa & Lin, 2014*). This is made possible through interconnecting tissues and a shared gastrovascular system (*Hemond, Kaluziak & Vollmer, 2014*). Individual polyps can thus act as cooperative systems, buffering negative microhabitat effects, colonising hostile areas, and permitting resource sharing should some polyps fail to capture food or light (*Murdock, 1978*; *Gateno et al., 1998*; *Bone & Keough, 2010*).

Additionally, intracolonial transport of essential resources via the gastrovascular system enables transference from established regions into zones of maximal energetic demand (*Taylor, 1977*). For example, organic products (lipids, glycerol, and glucose) have been shown to be translocated from established branch bases to growing tips, in order to contribute to calcification and tissue synthesis (*Fang, Chen & Chen, 1989*; *Oren, Rinkevich & Loya, 1997*). Established colony areas face less competitive interactions and are no longer undergoing rapid growth, and thus possess higher energy reserves (*Oku et al., 2002*). Meanwhile, growing regions with proliferating cells have higher metabolic rates and thus greater energetic demand (*Gladfelter, Michel & Sanfelici, 1989*; *Oku et al., 2002*).

Different colony regions can also be specialised for specific functions. For example, established colony regions are generally the most fecund, while the growing regions are often sexually sterile, and dedicated to growth and asexual reproduction (*Nozawa & Lin, 2014*). Such compartmentalisation of functional roles transcends to physiological processes. It is therefore reasonable to hypothesise that this, in turn, should manifest in zonation of nutritional resources and bioactive compounds, since different nutrients are required to fuel different life processes (*Sargent et al., 1999b*). For example, the quantity and nature of coral lipids and their constituent fatty acids (FA) varies significantly with photosynthesis, respiration, heterotrophy, cell replenishment, and reproduction (*Ward, 1995*; *Leuzinger, Anthony & Willis, 2003*; *Imbs, 2013*). In particular, high photosynthesis rates can be characterised by an abundance of carbon-rich compounds (*Muscatine, 1990*), while heterotrophically-derived nutrients are richer in nitrogen and phosphorus (*Houlbrèque & Ferrier-Pagès, 2009*). Additionally, protein fuels tissue growth, and calcification (*Conlan, Rocker & Francis, 2017*), as well as high metabolic rates in corals (*Oku et al., 2002*).

While numerous studies have examined intra-colonial variation in the biochemical composition along a single coral branch (i.e., branch base-tip) (*Taylor, 1977*; *Fang, Chen & Chen, 1989*; *Gladfelter, Michel & Sanfelici, 1989*; *Oku et al., 2002*; *Tang et al., 2015*), these results provide little insight into nutritional processes within the entire colony. To

date, comprehensive analyses expanding these findings to nutritional variation between colony regions (i.e., colony centre-edge) has yet to be conducted. Identifying nutritional compartmentalisation in coral colonies *in toto* would augment current understanding of the nutritional resources driving functional roles in healthy coral today, which is fundamental in informing coral reef research and management practices in the future.

Elucidating intracolonial variation in the nutritional profile may also facilitate an improved harvesting strategy of wild donor coral colonies, permitting judicious selection of branches with maximal survival, growth, and vigour prospects following fragmentation and translocation for reef rehabilitation efforts and captive husbandry (*Olivotto et al., 2011*; *Leal et al., 2016*). Increased global awareness of coral reef degradation has resulted in the development of rehabilitation strategies including fragmentation and translocation, to enable rapid regeneration of degraded reef sites (*Forsman et al., 2015*). However, fragmentation and translocation are highly stressful events for coral, and reports of successful cultivation from fragmented branches are sparse (*Arvedlund, Craggs & Pecorelli, 2003*; *Leal et al., 2016*).

Fragmentation causes tissue loss, lesions, and colony size reduction, which leads to an extensive and vulnerable recovery period (*Smith & Hughes, 1999*; *Lirman, 2000a*). This is exacerbated by translocation into foreign conditions, where water quality, temperature, light, and feed availability can vary significantly from the environment of origin. Importantly, the size and quality of initial energy reserves have been shown to significantly influence coral survivorship following stressful events (*Anthony et al., 2009*; *Sheridan et al., 2013*). As such, identifying the colony branch locations that possess 'optimal' nutritional profiles would be advantageous to buffer the stressors associated with fragmentation and translocation.

In addition, since coral colony size has been shown to influence several physiological aspects, including energy allocation to growth (*Anthony, Connolly & Willis, 2002*), reproduction (*Nozawa & Lin, 2014*), and primary production (*Jokiel & Morrissey, 1986*), the nutritional drivers behind these processes should also vary with colony size. As such, this study sought to test two hypotheses; (1) that coral branch biochemical composition will differ depending on its originating position within the colony, and (2) that colony size will influence the regional trends in branch biochemical composition. Here, we examined the nutritional composition along the longitudinal radius of two different size classes of a common, broadcast spawning scleractinian coral, *Acropora millepora*. The growth of *A. millepora* is characterised by radiate, skeletal accretion which manifests as multiple finger-like branches arrayed in roughly two dimensions.

## MATERIALS AND METHODS

### Sample collection

Sampling was conducted at depths of 2.6–5.4 m at Davies Reef in the Great Barrier Reef, Queensland, Australia (lat.: $-18°49.948'$S, long.:$147°37.995'$E) on the 9th of June, 2015. Ten colonies of two size classes of *Acropora millepora* were sampled (Field collections were approved by the Great Barrier Reef Marine Park Authority: G12/35236.1). The two size

(a)

L-centre    L-33    L-66    L-edge

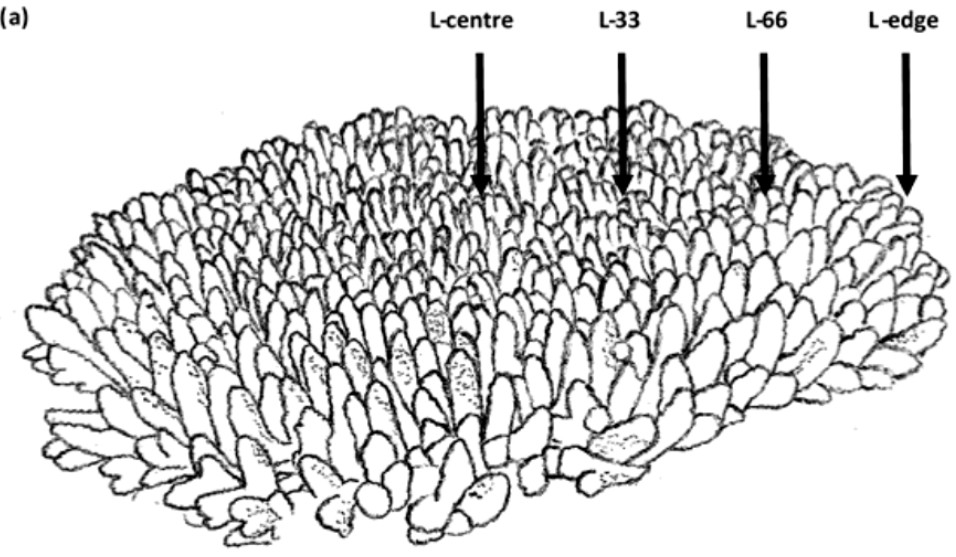

(b)

S-centre    S-50    S-edge

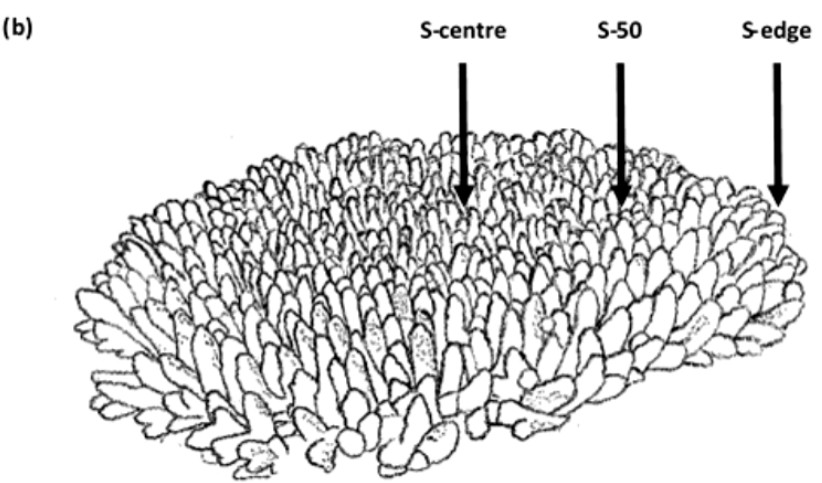

**Figure 1** **Branch sampling locations of *Acropora millepora* colonies.** (A) Larger colony ($4{,}376 \pm 741$ cm$^2$), L-centre: colony centre, L-33: 33.3% of the longitudinal radius length, L-66: 66.6% of the longitudinal radius length, L-edge: colony edge. (B) Smaller colony ($1{,}410 \pm 88$ cm$^2$), S-centre: colony centre, S-50: 50% of the longitudinal radius length, S-edge: colony edge (S-edge). For each size class, $n = 10$.

classes were larger: $4{,}376 \pm 741$ cm$^2$, and smaller: $1{,}410 \pm 88$ cm$^2$ (planar surface area), which represented two common size classes in the sampling area. For each colony, two replicate, adjacent branches were sampled from each location. For larger colonies, four locations were sampled: the centre of the colony (L-centre), 33.3% of the longitudinal radius length (L-33), 66.6% of the longitudinal radius length (L-66), and at the edge (L-edge) (Fig. 1A). For smaller colonies, branches were sampled at three locations: the centre of the colony (S-centre), 50% of the longitudinal radius length (S-50), and the edge of the colony (S-edge) (Fig. 1B).

## Zooxanthellae densities

Zooxanthellae were extracted using the air-spraying technique (*Szmant & Gassman, 1990*). Tissue was removed from the coral skeleton using a jet of high-pressure air from a hand gun (80 psi, 1 cm distance to coral). All sprayed tissue was captured in a thick, polyethylene bag containing 10 ml of ultrafiltered seawater (0.04 mm filtration). The tissue slurry was then poured from the plastic bag into a falcon tube, and the bag was double-rinsed with ultrafiltered seawater that was also collected. The slurry was then homogenised for 20 s (Ultra-Turrax T10B; IKA Labortechnik, Staufenim Breisgau, Germany). A 500 uL aliquot of tissue slurry was taken and combined with a 500 uL aliquot of 3% formalin: filtered seawater for preservation. Zooxanthellae were counted in triplicate using a haemocytometer. Zooxanthellae densities were then standardised to branch surface area ($cm^2$), which was obtained through the simple geometry technique, which has been shown to be suitable for *Acropora* spp. (*Naumann et al., 2009*), using the software Fiji ImageJ (*Schindelin et al., 2012*).

## Proximate analysis

Proximate analysis refers to the measurement of the relative amounts protein, lipid, moisture, ash, and carbohydrate in an organism. Once the zooxanthellae aliquot was taken, the tissue slurry and denuded skeletons were freeze-dried for 72 hrs. Following freeze-drying, the denuded skeletons were placed inside a stainless steel mortar and pestle (cleaned with methanol), which was placed inside a manual laboratory hydraulic press (Model C; Fred S. Carver Inc., Summit, NJ, USA), and pressurised to 70 kN, crushing the corals to a fine powder. This powder was then recombined with the dried tissue to account for organic material present in both the coral skeleton and tissue (*Conlan, Rocker & Francis, 2017*). The recombined coral powder was then extracted for total lipid content according to the method described in *Conlan et al. (2014)*. Dry samples were weighed then soaked overnight in a 3 mL aliquot of dichloromethane: methanol ($CH_2Cl_2$:$CH_3OH$). The following morning, this mixture was filtered and the solid residue re-suspended and soaked for a further 10 min with another 3 mL aliquot of $CH_2Cl_2$:$CH_3OH$, followed by a further filtration step. This process was repeated three times. The combined filtrates ($\sim$9 mL) were then transferred into a separation funnel and combined with a 4.5 mL sample washing solution of KCl (0.44%) in $H_2O$/$CH_3OH$ (3:1). The mixture was shaken and allowed to settle overnight. The following morning, the bottom layer containing the extracted lipid was recovered and the solvent was evaporated under nitrogen. The lipid content was then quantified to four decimal places. Protein content was determined according to the Kjeldahl method (crude protein calculated as nitrogen $\times$ 6.25) in an automated Kjeltech (Tecator, Sweden). Total ash was determined by incineration in a muffle furnace (Model WIT; C & L Fetlow, Blackburn, Victoria, Australia) at 450 °C for 12 h. The ash content was subtracted from the total composition to obtain ash free dry weight (AFDW), which excludes the inorganic component. Carbohydrate/ nitrogen free extract (NFE) was obtained

by subtracting the ash, lipid, and protein from the total sample mass. Energy was calculated using the combustion enthalpies for lipid (39.5 J mg$^{-1}$) and protein (23.9 J mg$^{-1}$) from *Gnaiger & Bitterlich (1984)*.

### Lipid class and fatty acid analysis

Lipid class analysis was determined using an Iatroscan MK 6s thin layer chromatography-flame ionisation detector (Mitsubishi Chemical Medience, Tokyo Japan) according to the method of *Conlan et al. (2014)*. Each sample was spotted in duplicate on silica gel S5-chromarods (5 μm particle size) with lipid separation following a two-step elution sequence: (1) phosphatidylethanolamine (PE), phosphatidylcholine (PC) and lysophosphatidylcholine (LPC) elution was achieved in a dichloromethane/methanol/water (50:20:2, by volume) solvent system run to half height (∼15 min); and (2) after air drying, wax ester (WAX), triacylglycerol (TAG), free fatty acid (FFA), 1,2-diacylglycerol (1,2DAG), and sterol (ST) elution was achieved in a hexane/diethyl ether/formic acid (60:15:1.5, by volume) solvent system run to full height (∼30 min). Since glycolipids commonly elute with monoacylglycerols and pigments, including chlorophyll, the term "acetone mobile polar lipid" (AMPL) was used in the present study (*Parrish, Bodennec & Gentien, 1996*). AMPL was quantified using the 1-monopalmitoyl glycerol standard (Sigma-Aldrich Co., USA), which has demonstrated a response that is intermediate between glycoglycerolipids and pigments (*Parrish, Bodennec & Gentien, 1996*).

Following initial lipid extraction, FA were esterified into methyl esters using an acid-catalysed methylation method and then analysed by gas chromatography as described in *Conlan et al. (2014)*.

### Statistical analysis

Data were analysed using R software version 2.3.1 (*RStudio Team, 2015*; *R Development Core Team, 2016*). Due to non-normality and heteroscedasticity (detected via Shapiro–Wilk and Levene's tests, respectively), as well as some negative values, data were transformed using a Yeo-Johnson power transformation (*caret* package (*Kuhn, 2016*)). Transformed data were then analysed using a one-way analysis of variance (ANOVA) for each parameter measured. Where statistical differences were detected, a TukeyHSD *post-hoc* test was employed at a significance level of $P < 0.05$ (*agricolae* package (*De Meniburu, 2015*)). FA profiles (mg g lipid$^{-1}$) were also analysed using a linear discriminant analysis (LDA), in order to visualize the relationships between colony position (*MASS* package (*Venables & Ripley, 2002*)). An LDA biplot was included to show the top fifteen FA driving the differences between colony position. The colour gradient of the vectors show percentage contribution to LDA loadings. Ellipses show 95% confidence intervals. Figures were prepared using the *ggplot2* package (*Wickham, 2009*).

## RESULTS

### Proximate composition and zooxanthellae density

For the larger colonies, the zooxanthellae densities at the edge were lower than the next immediate position, L-66 (cells cm$^{2-1}$) (Fig. 2A), although this was not statistically

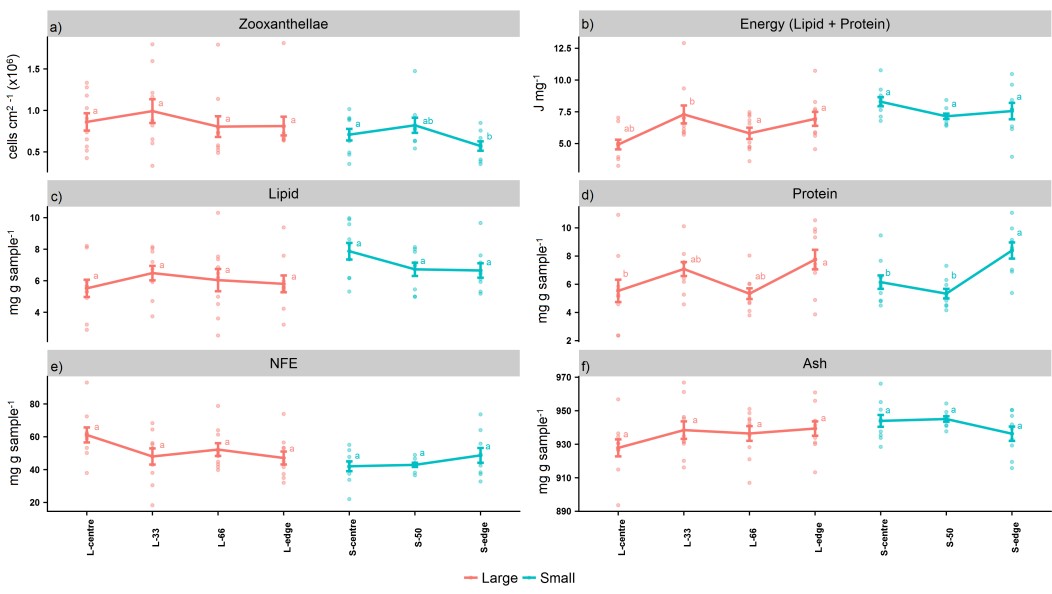

**Figure 2** **Proximate composition, energy content, and zooxanthellae density of distal intracolonial locations in two size classes of *Acropora millepora* colonies.** (A) Total lipid concentration, (B) Total protein concentration, (C) Total nitrogen-free extract concentration (NFE), (D) Total ash concentration, (E) Zooxanthellae density, (F) Energy content (using caloric enthalpies of lipid + protein). L-centre: larger colony centre, L-33: 33.3% of the longitudinal radius length, L-66: 66.6% of the longitudinal radius length, L-edge: larger colony edge, S-centre: smaller colony centre, S-50: 50% of the longitudinal radius length, S-edge: smaller colony edge (S-edge). Values are presented as means ± SEM. Letters in common denote no significant difference ($P < 0.05$). For each size class, $n = 10$.

significant ($P_{\text{ANOVA}} > 0.05$, $P_{\text{HSD}} > 0.05$). On the other hand, the smaller colonies recorded significantly lower zooxanthellae densities at the edge in comparison to the centre ($P_{\text{ANOVA}} < 0.05$, $P_{\text{HSD}} < 0.05$).

The caloric enthalpies showed that, for both size classes, the edge and next immediate position contained the highest energy contents, while the S-centre, L-centre, and L-33 positions were low (Fig. 2B). Although these differences were not significant for the smaller colonies ($P_{\text{ANOVA}} > 0.05$, $P_{\text{HSD}} > 0.05$), for the larger colonies, the L-33 position was significantly lower compared to the L-66 position ($P_{\text{ANOVA}} < 0.01$, $P_{\text{HSD}} < 0.01$) as well as the edge ($P_{\text{ANOVA}} < 0.01$, $P_{\text{HSD}} < 0.05$).

Within the size classes, there were no significant differences in total lipid ($P_{\text{ANOVA}} > 0.05$, $P_{\text{HSD}} > 0.05$) (mg g sample$^{-1}$) (Fig. 2C), although both edge positions contained lower lipid compared to the next, immediate position. Within both size classes, there was a progressive increase in protein from the colony centre toward the edge (mg g sample$^{-1}$) (Fig. 2D). For the larger colonies, protein at the edge was significantly higher than the centre ($P_{\text{ANOVA}} < 0.05$, $P_{\text{HSD}} < 0.05$), while for the smaller colonies, the edge was significantly higher than the centre and S-50 positions ($P_{\text{ANOVA}} < 0.01$, $P_{\text{HSD}} < 0.01$).

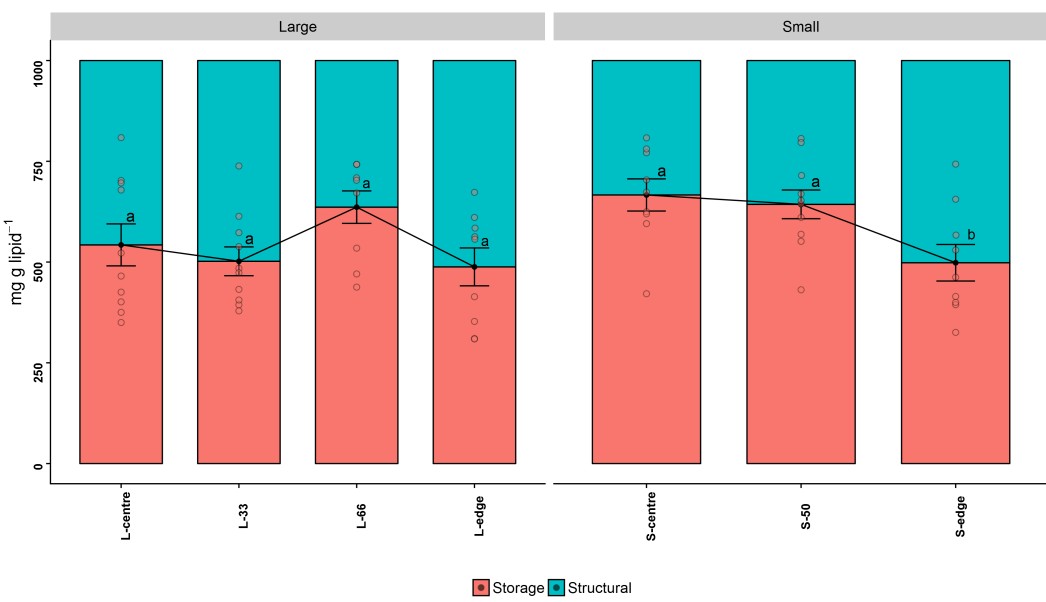

**Figure 3** **Lipid class composition - relative proportion of storage and structural lipids of distal intra-colonial locations in two size classes of *Acropora millepora* colonies.** L-centre: larger colony centre, L-33: 33.3% of the longitudinal radius length, L-66: 66.6% of the longitudinal radius length, L-edge: larger colony edge, S-centre: smaller colony centre, S-50: 50% of the longitudinal radius length, S-edge: smaller colony edge (S-edge). Values are presented as means ± SEM. Letters in common denote no significant difference ($P < 0.05$). For each size class, $n = 10$.

## Lipid class composition

Storage lipids consist of the classes WAX, TAG, FFA, and 1,2-DAG, while structural lipids consist of ST, AMPL, PE, PS-PI, and PC. The storage lipid proportion in the larger colonies followed a similar trend to the total lipid concentration, with the L-66 position containing the highest storage lipids, although this was not statistically significant (mg g lipid$^{-1}$) ($P_{ANOVA} > 0.05$, $P_{HSD} > 0.05$) (Fig. 3A). Similarly, in the smaller colonies, the edge recorded the lowest storage lipid concentration, and this was significant compared to the S-centre and S-50 positions ($P_{ANOVA} < 0.05$, $P_{HSD} < 0.05$) (Fig. 3B, Table 1).

Neither size class recorded significant differences in TAG concentration between sites ($P_{ANOVA} > 0.05$, $P_{HSD} > 0.05$). However, the highest TAG concentration in the larger colonies was recorded at L-66, and at the centre position for the smaller colonies. AMPL was significantly higher at S-edge in comparison to the centre and S-50 ($P_{ANOVA} < 0.05$, $P_{HSD} < 0.05$). For the larger colonies, the highest AMPL concentrations were found at the edge and L-33 positions, which were significantly higher than the L-66 ($P_{ANOVA} < 0.05$, $P_{HSD} < 0.05$).

## Fatty acid and fatty alcohol composition

In both size classes, the edge positions contained the lowest total FA concentration (S-edge: 347 ± 58.6 mg g lipid$^{-1}$ and L-edge: 382 ± 52.6 mg g lipid$^{-1}$) ($P_{ANOVA} < 0.01$) (Table 2). In the smaller colonies, the highest total FA were found in the centre (519 ± 75.1 mg g lipid$^{-1}$), which was significantly higher compared to the edge ($P_{HSD} < 0.01$). The larger

**Table 1  Lipid class composition (mg g lipid$^{-1}$) of distal intracolonial locations in two size classes of *Acropora millepora* colonies.** L-centre: large colony centre, L-33: 33.3% of the longitudinal radius length, L-66: 66.6% of the longitudinal radius length, L-edge: large colony edge, S-centre: small colony centre, S-50: 50% of the longitudinal radius length, S-edge: small colony edge (S-edge). Values are presented as means ± SEM. For each colony size, values in the same row size that do not share the same superscripts are significantly different ($P < 0.05$).

| Lipid class (mg g lipid$^{-1}$) | Small | | | Large | | | |
|---|---|---|---|---|---|---|---|
| | Centre | S-50 | Edge | Centre | L-33 | L-66 | Edge |
| Wax ester | 114 ± 15.9[b] | 171 ± 48.5[a] | 118 ± 15.8[ab] | 194 ± 92.3[a] | 130 ± 7.71[a] | 125 ± 9.88[a] | 119 ± 6.77[a] |
| Triacylglycerol | 499 ± 94.1[a] | 411 ± 92.8[a] | 311 ± 94.9[a] | 270 ± 123[a] | 301 ± 69.9[a] | 456 ± 69.2[a] | 312 ± 75.3[ab] |
| Free fatty acid | 14.6 ± 3.48[a] | 16.5 ± 5.78[a] | 19 ± 3.29[a] | 20.1 ± 7.05[a] | 18.3 ± 2.73[a] | 17 ± 3.16[a] | 22.2 ± 2.08[a] |
| 1,2-diacylglycerol | 39.1 ± 13[a] | 44.5 ± 13[a] | 50.6 ± 14.5[a] | 58.5 ± 19.4[a] | 52.7 ± 22.1[a] | 38 ± 10.7[a] | 34.5 ± 14[a] |
| Sterol | 44.9 ± 9.17[b] | 48.5 ± 8.31[b] | 71.9 ± 9.47[a] | 56.4 ± 9.42[a] | 66 ± 7.72[a] | 63.9 ± 28[a] | 74.1 ± 6.81[a] |
| AMPL | 96.7 ± 23[b] | 100 ± 20.6[b] | 150 ± 25.6[a] | 118 ± 21.3[ab] | 138 ± 17[a] | 88.5 ± 12.2[b] | 145 ± 27[a] |
| Phosphatidylethanolamine | 49.9 ± 9.51[b] | 54.7 ± 9.57[ab] | 69.3 ± 12.9[a] | 70.5 ± 14.1[a] | 68.9 ± 17.5[a] | 54.5 ± 8.15[a] | 73.4 ± 9.82[a] |
| Phosphatidylserine/phosphatidylinositol | 53.1 ± 16.7[b] | 69 ± 20[ab] | 89.6 ± 15.2[a] | 89.7 ± 25.7[a] | 95.1 ± 22.3[a] | 69.1 ± 12.9[a] | 92.9 ± 16.5[a] |
| Phosphatidylcholine | 77.3 ± 19.2[b] | 81.1 ± 13.6[ab] | 115 ± 20.3[a] | 103 ± 19.4[a] | 116 ± 15.3[a] | 72.6 ± 7.72[b] | 121 ± 16.5[a] |
| Lysophosphatidylcholine | 11.7 ± 11.2[a] | 3.12 ± 5.7[a] | 6.05 ± 10.5[a] | 19.3 ± 14.5[a] | 13.5 ± 14.8[a] | 15.1 ± 12[a] | 6.1 ± 9.15[a] |
| $\sum$Storage | 66.6 ± 6.91[a] | 64.3 ± 6.5[a] | 49.8 ± 7.85[b] | 54.3 ± 8.23[a] | 50.2 ± 5.65[a] | 63.6 ± 6.05[a] | 48.8 ± 7.05[a] |
| $\sum$Structural | 33.4 ± 6.91[b] | 35.7 ± 6.5[b] | 50.2 ± 7.85[a] | 45.7 ± 8.23[a] | 49.8 ± 5.65[a] | 36.4 ± 6.05[a] | 51.2 ± 7.05[a] |
| Storage:Structural | 2.34 ± 0.66[a] | 2.1 ± 0.64[ab] | 1.17 ± 0.45[b] | 1.55 ± 0.59[a] | 1.14 ± 0.33[a] | 1.99 ± 0.42[a] | 1.09 ± 0.29[a] |

**Table 2  Fatty acid and fatty alcohol composition (mg g lipid$^{-1}$) of distal intracolonial locations in two size classes of *Acropora millepora* colonies.** L-centre: large colony centre, L-33: 33.3% of the longitudinal radius length, L-66: 66.6% of the longitudinal radius length, L-edge: large colony edge, S-centre: small colony centre, S-50: 50% of the longitudinal radius length, S-edge: small colony edge (S-edge). Values are presented as means ± SEM. For each colony size, values in the same row size that do not share the same superscripts are significantly different ($P < 0.05$).

| Fatty acids (mg g lipid$^{-1}$) | Small | | | Large | | | |
|---|---|---|---|---|---|---|---|
| | Centre | S-50 | Edge | Centre | L-33 | L-66 | Edge |
| 14:0 | 25.2 ± 4.32[a] | 21.9 ± 2.71[a] | 14.4 ± 3.14[b] | 21.2 ± 3.95[ab] | 17.8 ± 2.97[ab] | 25.8 ± 3.97[a] | 14.9 ± 2.52[b] |
| 16:0 | 224 ± 38.6[a] | 192 ± 22[ab] | 134 ± 29.4[b] | 188 ± 32.2[ab] | 159 ± 23.4[b] | 238 ± 32.4[a] | 144 ± 26.2[b] |
| 18:0 | 38.1 ± 6.19[a] | 33.6 ± 4.22[a] | 33.5 ± 3.88[a] | 30.8 ± 3.11[a] | 29.4 ± 3.64[a] | 33.6 ± 3.66[a] | 33.1 ± 3.76[a] |
| $\sum$SFA | 296 ± 48.9[a] | 255 ± 28[ab] | 189 ± 35.9[b] | 248 ± 37.5[ab] | 214 ± 28.2[b] | 306 ± 38.9[a] | 199 ± 31.3[b] |
| 16:0-OH | 68.4 ± 13[a] | 59.8 ± 9.22[a] | 37.7 ± 9.63[b] | 55.1 ± 11.7[ab] | 46.9 ± 8.75[b] | 75.8 ± 10.2[a] | 44 ± 9.84[b] |
| 18:1n-9 | 18.4 ± 3.41[a] | 16.4 ± 2.42[a] | 11.1 ± 2.72[b] | 19.5 ± 3.76[ab] | 15.8 ± 1.97[ab] | 24.4 ± 3.71[a] | 14.2 ± 2.93[b] |
| 20:1n-11 | 14.3 ± 1.9[a] | 11.8 ± 1.23[ab] | 9.53 ± 1.36[b] | 12.8 ± 1.48[a] | 11.8 ± 1.37[a] | 14.6 ± 1.74[a] | 11.1 ± 1.2[a] |
| $\sum$MUFA | 51.2 ± 6.99[a] | 44.5 ± 4.44[ab] | 32.8 ± 5.73[b] | 48.8 ± 7.54[ab] | 41.9 ± 4.75[b] | 59.4 ± 7.49[a] | 38.7 ± 5.7[b] |
| 18:3n-6 | 24.8 ± 4.38[a] | 21.2 ± 2.34[a] | 13.4 ± 3.18[b] | 20.1 ± 3.87[ab] | 17 ± 2.3[b] | 26.6 ± 3.48[a] | 15.7 ± 3.28[b] |
| 20:4n-6 | 8.22 ± 1.4[b] | 8.3 ± 1.63[ab] | 11.2 ± 1.26[a] | 8.39 ± 1.6[b] | 8.79 ± 1.34[b] | 8.13 ± 0.81[b] | 12.5 ± 1.64[a] |
| 20:5n-3 | 21.4 ± 1.85[a] | 19.6 ± 2.72[a] | 21.7 ± 2.76[a] | 19.1 ± 1.5[a] | 19.7 ± 2.15[a] | 20.9 ± 1.93[a] | 23.4 ± 2.1[a] |
| 22:6n-3 | 16.9 ± 3.02[a] | 14.5 ± 2.08[ab] | 11.1 ± 2.29[b] | 13.6 ± 2.4[a] | 11.9 ± 1.88[a] | 17 ± 2.47[a] | 12.3 ± 1.81[a] |
| $\sum$PUFA | 164 ± 21[a] | 146 ± 15.9[ab] | 119 ± 17.6[b] | 142 ± 16.5[ab] | 128 ± 14[b] | 175 ± 19.5[a] | 136 ± 16.2[b] |
| $\sum$Fatty alcohol | 75.4 ± 13.3[a] | 66.5 ± 9.66[ab] | 44.6 ± 10.2[b] | 61.4 ± 11.9[b] | 53.1 ± 8.79[b] | 82.9 ± 10.8[a] | 51.7 ± 10.1[b] |
| Total | 519 ± 75.1[a] | 452 ± 46.8[ab] | 347 ± 58.6[b] | 445 ± 61.3[ab] | 391 ± 46.1[b] | 547 ± 66[a] | 382 ± 52.6[b] |
| $\sum$n-3 PUFA | 44.9 ± 4.65[a] | 40.2 ± 5.04[a] | 39.3 ± 5.16[a] | 38.8 ± 3.31[a] | 37.8 ± 4.04[a] | 44.5 ± 4.94[a] | 43.1 ± 3.93[a] |
| $\sum$n-6 PUFA | 26.2 ± 2.27[a] | 24.9 ± 3.38[a] | 28.1 ± 2.53[a] | 28.2 ± 3.27[a] | 26.5 ± 3.1[a] | 27.7 ± 2.36[a] | 33.1 ± 3.06[a] |

**Table 3** Fatty acid and fatty alcohol composition (% fatty acids) of distal intracolonial locations in two size classes of *Acropora millepora* colonies. L-centre: large colony centre, L-33: 33.3% of the longitudinal radius length, L-66: 66.6% of the longitudinal radius length, L-edge: large colony edge, S-centre: small colony centre, S-50: 50% of the longitudinal radius length, S-edge: small colony edge (S-edge). Values are presented as means ± SEM. For each colony size, values in the same row size that do not share the same superscripts are significantly different ($P < 0.05$).

| Fatty acids (% fatty acids) | Small | | | Large | | | |
|---|---|---|---|---|---|---|---|
| | Centre | S-50 | Edge | Centre | L-33 | L-66 | Edge |
| 14:0 | 4.8 ± 0.16[a] | 4.83 ± 0.22[a] | 4.05 ± 0.29[b] | 4.66 ± 0.29[a] | 4.49 ± 0.31[a] | 4.67 ± 0.26[a] | 3.83 ± 0.21[b] |
| 16:0 | 42.6 ± 1.68[a] | 42.3 ± 1.74[a] | 37.5 ± 2.69[b] | 41.4 ± 1.68[a] | 40.4 ± 1.46[ab] | 43.2 ± 0.85[a] | 36.8 ± 2.2[b] |
| 18:0 | 7.43 ± 0.85[b] | 7.45 ± 0.63[b] | 10 ± 1.05[a] | 7.27 ± 0.92[ab] | 7.66 ± 0.77[ab] | 6.23 ± 0.56[b] | 8.92 ± 0.8[a] |
| $\sum$SFA | 56.6 ± 1.39[a] | 56.3 ± 1.52[a] | 53.8 ± 1.79[a] | 55.2 ± 1.17[a] | 54.6 ± 1.13[ab] | 55.6 ± 0.79[a] | 51.6 ± 1.74[b] |
| 16:0-OH | 13 ± 0.76[a] | 13.1 ± 0.88[a] | 10.4 ± 1.13[b] | 12 ± 1.12[ab] | 11.8 ± 1.11[ab] | 13.8 ± 0.5[a] | 11.1 ± 1.18[b] |
| 18:1n-9 | 3.59 ± 0.62[a] | 3.67 ± 0.47[a] | 3.17 ± 0.51[a] | 4.27 ± 0.36[a] | 4.09 ± 0.35[a] | 4.41 ± 0.27[a] | 3.7 ± 0.41[a] |
| 20:1n-11 | 2.77 ± 0.17[a] | 2.63 ± 0.23[a] | 2.81 ± 0.23[a] | 2.93 ± 0.25[a] | 3.07 ± 0.26[a] | 2.7 ± 0.18[a] | 2.99 ± 0.25[a] |
| $\sum$MUFA | 9.97 ± 0.95[a] | 9.92 ± 0.68[a] | 9.49 ± 0.71[a] | 10.9 ± 0.35[a] | 10.8 ± 0.4[a] | 10.8 ± 0.23[a] | 10.1 ± 0.4[a] |
| 18:3n-6 | 4.31 ± 0.68[a] | 4.39 ± 0.61[a] | 6.59 ± 1.14[b] | 4.57 ± 0.64[ab] | 5.18 ± 0.62[a] | 3.91 ± 0.33[ab] | 6.47 ± 0.87[b] |
| 20:4n-6 | 3.23 ± 0.24[b] | 3.17 ± 0.26[b] | 3.14 ± 0.16[a] | 3.01 ± 0.2[b] | 2.99 ± 0.2[ab] | 3.08 ± 0.13[b] | 3.23 ± 0.2[a] |
| 20:5n-3 | 4.73 ± 0.44[b] | 4.72 ± 0.31[b] | 3.76 ± 0.43[a] | 4.4 ± 0.32[b] | 4.32 ± 0.16[ab] | 4.85 ± 0.2[b] | 4.01 ± 0.36[a] |
| 22:6n-3 | 1.76 ± 0.55[a] | 1.88 ± 0.44[a] | 3.52 ± 0.76[a] | 2.13 ± 0.59[a] | 2.37 ± 0.45[a] | 1.55 ± 0.23[a] | 3.54 ± 0.7[a] |
| $\sum$PUFA | 19 ± 1.67[b] | 19.2 ± 1.52[b] | 24.2 ± 2.54[a] | 20.5 ± 1.84[b] | 21.2 ± 1.72[ab] | 18.3 ± 0.77[b] | 25 ± 2.36[a] |
| $\sum$Fatty alcohol | 14.4 ± 0.62[a] | 14.5 ± 0.78[a] | 12.5 ± 0.91[b] | 13.4 ± 1.01[a] | 13.4 ± 0.98[a] | 15.1 ± 0.49[a] | 13.2 ± 0.95[a] |
| Total *(% lipid)* | 51.9 ± 7.5[a] | 45.2 ± 4.7[a] | 34.7 ± 5.9[b] | 44.5 ± 6.1[ab] | 39.1 ± 4.6[b] | 54.7 ± 6.6[a] | 38.2 ± 5.2[b] |
| $\sum$n-3 PUFA | 8.87 ± 0.86[b] | 8.92 ± 0.91[b] | 11.7 ± 1.43[a] | 9.03 ± 0.74[b] | 9.81 ± 0.8[ab] | 8.21 ± 0.42[b] | 11.7 ± 1.26[a] |
| $\sum$n-6 PUFA | 5.39 ± 1.07[b] | 5.57 ± 0.84[b] | 8.67 ± 1.46[a] | 7 ± 1.47[ab] | 7.04 ± 1.02[ab] | 5.2 ± 0.48[b] | 9.21 ± 1.4[a] |

colonies contained the highest total FA in the L-66 position ($547 \pm 66$ mg g lipid$^{-1}$), which was significantly higher than the L-33 and edge positions ($P_{HSD} < 0.05$). These trends extended to the individual FA (mg g lipid$^{-1}$). Notably, DHA showed a distal decrease from the centre to the edge in the smaller colonies (S-centre: $16.9 \pm 3.02$ mg g lipid$^{-1}$–S-edge: $11.1 \pm 2.29$ mg g lipid$^{-1}$) ($P_{ANOVA} < 0.05$, $P_{HSD} < 0.05$), yet in the larger colonies was found in the highest concentrations in the L-66 position ($17 \pm 2.47$ mg g lipid$^{-1}$), and the lowest in the L-33 and edge positions ($\sim 12$ mg g lipid$^{-1}$) ($P_{ANOVA} > 0.05$, $P_{HSD} > 0.05$). This was mirrored in the total PUFA concentrations for both size classes. ARA was an exception, being present in the highest concentrations at the edge, regardless of size class ($P_{ANOVA} < 0.05$).

When viewed as % FA, the edge positions recorded significantly higher PUFA concentrations compared to all other positions for the smaller colonies ($P_{ANOVA} < 0.01$, $P_{HSD} < 0.01$), and compared to the centre and L-66 position for the larger colonies (Table 3) ($P_{ANOVA} < 0.01$, $P_{HSD} < 0.05$). The edge positions also contained the lowest saturated fatty acids (SFA) concentrations, and this was significant compared to the centre and L-66 positions for the larger colonies ($P_{ANOVA} < 0.01$, $P_{HSD} < 0.05$).

In the LDA of the FA composition (mg g lipid$^{-1}$), the first two linear discriminates explained 83.3% of the total variation between size classes and colony positions (Fig. 4A), with an established Wilks value of 0.0004. The analysis showed a clear difference between the smaller and larger colonies, with the smaller colonies grouping mostly on the positive

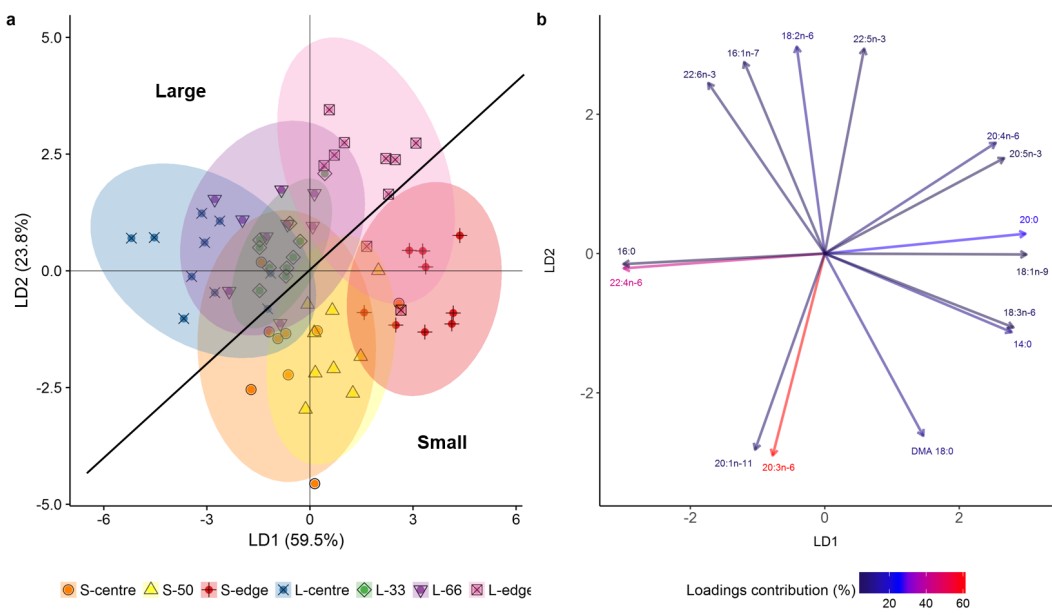

**Figure 4** **Linear discriminant analysis (LDA) (A) score plot and (B) biplot, showing overall fatty acid profile (mg g lipid$^{-1}$) of distal intracolonial locations in two size classes of *Acropora millepora* colonies.** L-centre: larger colony centre, L-33: 33.3% of the longitudinal radius length, L-66: 66.6% of the longitudinal radius length, L-edge: larger colony edge, S-centre: smaller colony centre, S-50: 50% of the longitudinal radius length, S-edge: smaller colony edge (S-edge). (A) Ellipses show 95% confidence intervals for each treatment. (B) Vectors show top fifteen individual fatty acids and fatty alcohols contributing to the overall variance between treatments. Colour gradient shows percentage contribution to LDA loadings. For each size class, $n = 10$.

side of LD1 and negative side of LD2, and the larger showing the inverse. Despite this separation, both size classes followed the same trends. There was clear separation of the edge positions from centre positions. The middle regions (S-50, L-33, L-66) largely grouped between their respective centre and edge positions. The overlapping of the S-centre and S-50 groups illustrates the similarity between these positions, as well as for the L-33, L-66, and L-centre positions.

The LDA biplot (Fig. 4B) shows that the size class separation was largely due to 18 DMA for the small colonies, while 22:6n-3 (DHA), 16:1n-7, and 18:n-6 drove the larger colony separation. Separation of the two edge positions was strongly influenced by the polyunsaturated fatty acids (PUFA) 18:3n-6, 20:5n-3 (EPA), and 20:4n-6 (ARA) as well as the SFA 14:0 and 20:0, and the monounsaturated fatty acid (MUFA), 18:1n-9. Meanwhile, the centre positions were largely influenced by 16:0, 22:4n-6, 20:1n-11, and 20:3n-6.

## DISCUSSION

The present study examined the biochemical composition along the longitudinal radius of *A. millepora* colonies, providing a unique account of nutritional resource compartmentalisation stemming from branch position and functional specialisation within a single coral colony. These findings provide new insights into colony-wide zonation of the

comprehensive nutritional profile from a length-wise (i.e., colony centre-edge) perspective since, until now, previous studies have been limited to individual coral branches from a height-wise (i.e., branch base-tip) perspective (*Taylor, 1977*; *Fang, Chen & Chen, 1989*; *Gladfelter, Michel & Sanfelici, 1989*; *Oku et al., 2002*; *Tang et al., 2015*).

## Total protein concentration and zooxanthellae densities

Since growth of branching scleractinian corals is characterised by radiate skeletal accretion, the colony edges generally undergo the greatest growth rates (*Kaandorp, 1995*). This can account for the significant, distal increase in total protein from the centre regions toward the edge recorded in both colony size classes. High protein levels are characteristic of actively growing sites, where protein synthesis and retention facilitates calcification, tissue and polyp production (*Mitterer, 1978*; *Conlan, Rocker & Francis, 2017*). These results agree well with previous studies on the nutritional profile of individual soft coral branches (base-tip), which showed increases in protein concentration at the actively growing tip, compared to the established base (*Tentori & Allemand, 2006*; *Tentori & Thomson, 2009*). However, these results show that when individual branches are analysed as a whole, the mean protein concentrations of each branch position manifests in a nutritional gradient from the colony centre toward the edge that is parallel to that of an individual branch from the base toward the tip. This indicates that a coral colony exhibits similar nutritional gradients in two dimensions; height-wise (i.e., base-tip) and length-wise (i.e., centre-edge), and that the latter trend is manifestly stronger than the former.

The significantly lower zooxanthellae densities recorded at the smaller colony edges is again comparable to growing branch tips, where newly formed tissues are yet to be fully colonised by symbionts (*Goreau, 1959*; *Gladfelter, Michel & Sanfelici, 1989*), further demonstrating the similarities between height-wise and length-wise gradients within a coral colony. In contrast, the lowest zooxanthellae densities in the larger colonies were akin to the most dense region in the smaller colonies, possibly due to their larger size, as larger coral colonies are suggested to undergo more rapid zooxanthellae proliferation compared to smaller colonies (*Muscatine, Mccloskey & Loya, 1985*).

## Total lipid, lipid classes, and fatty acids

Although zooxanthellae densities are directly correlated with lipid concentrations in corals (*Fang, Chen & Chen, 1989*), the lower density at the colony edges of smaller corals did not correlate with low total lipid concentrations. This may suggest an increase in heterotrophic feeding by the edge regions to supplement the reduced phototrophic nutrient supply (*Lesser, 2012*; *Levas et al., 2015*). The increased protein concentrations at the edges support this, since exogenous food is known to supply nitrogen-rich building blocks needed for tissue and skeletal biosynthesis (*Osinga et al., 2011*). Alternatively, this may indicate intracolonial lipid translocation from the established, zooxanthellae-dense regions toward the colony edge (*Fang, Chen & Chen, 1989*). Inner colony regions generally acquire an energy surplus, since photosynthesis greatly exceeds respiration and competitive interactions are reduced (*Gladfelter, Michel & Sanfelici, 1989*; *Lirman, 2000a*). As such, energy accumulated by the inner regions can be mobilised via the coral's gastrovascular system and concentrated in

zones of maximal energetic demand, such as the growing edge (*Oren, Brickner & Loya, 1998*; *Bone & Keough, 2010*; *Marfenin, 2015*).

Within the total lipid concentration, storage lipids (WAX, TAG, FFA, and 1,2-DAG) are generally associated with energy supply, while structural lipids (ST, AMPL, PE, PS-PI, and PC) are important cell membrane constituents, facilitating cell membrane fluidity and permeability, and performing vital cell signalling processes (*Lee, Hagen & Kattner, 2006*; *Ferrier-Pagès et al., 2016*). As such, the significantly lower storage lipid concentrations at the edges of the smaller colonies may reflect lipid catabolism to supply the energy demanded by proliferating cells (*Oku et al., 2002*; *Denis et al., 2013*). Concurrently, the increased structural lipid proportion likely reflects tissue synthesis, since ST and phospholipids constitute the building blocks of cell membranes (*Imbs et al., 2010*).

In contrast to the smaller colonies, there were no significant differences in storage lipid concentrations between positions for the larger colonies. This may suggest a shift in inner region function of the larger colonies, such that they have transitioned from nutrient assimilation to nutrient remobilisation—allowing the rest of the colony to benefit from their accumulated reserves, as has been shown in colonial plants (*Avila-Ospina et al., 2014*). Interestingly, although the combustion enthalpy for protein (23.9 J mg$^{-1}$) is far lower than lipid (39.5 J mg$^{-1}$) (*Gnaiger & Bitterlich, 1984*), the high protein at the edges manifested in energy levels similar to the positions containing the maximum lipid concentrations, demonstrating the substantial allocation of metabolic resources to colony growth (*Ramos-Silva et al., 2014*).

The trends between the smaller and larger colonies extended to the total FA concentrations (mg g lipid$^{-1}$) whereby significantly higher concentrations were found in the centre position for the smaller colonies, and the L-66 position for the larger colonies. This conforms to the lipid class results in these positions, which contained higher TAG concentrations, which comprises three esterified FA, while the edge is higher in phospholipids and ST, which possess two or less esterified FA (*Lee, Hagen & Kattner, 2006*). The total FA trends were reflected in most individual FA concentrations and indicate that these positions, along with the edges, represent the major energy sinks in their respective colony sizes.

Notably, the S-centre, S-50, and L-66 positions exhibited significantly higher SFA and MUFA concentrations, largely due to 14:0, 16:0, and 18:1n-9. These FA are known to be energy-rich and readily catabolised, and are largely derived from zooxanthellae, corresponding to the high zooxanthellae densities at these positions (*Figueiredo et al., 2012*). Furthermore, large reserves of energy-rich lipid constituents, including TAG, SFA, and MUFA, are typically associated with coral reproduction; an energetically expensive life process (*Figueiredo et al., 2012*). Since corals must partition metabolic energy amongst several important physiological processes (*Leuzinger, Willis & Anthony, 2012*), the predominance of these compounds in the L-66, S-centre, and S-50 positions implicate these regions as being reproductively active. This correlates with the findings of *Nozawa & Lin (2014)*, who showed that polyps obtained from the L-50 position of large *Acropora hyacinthus* colonies had the highest fecundity levels, while the highest fecundity was detected in the centre and S-50 positions of smaller colonies.

In contrast, the edge positions were conspicuously low in these readily catabolised materials, conforming to both the lower zooxanthellae densities and the well-documented sterility of actively growing regions in *Acropora* colonies (*Hemond, Kaluziak & Vollmer, 2014*; *Nozawa & Lin, 2014*). Correspondingly, the edges contained significantly higher PUFA proportions (% FA), which are known to be major cell membrane constituents and are required for high growth and development rates (*Brett & Muller-Navara, 1997*; *Sargent et al., 1999a*).

**Considerations for coral harvesting**

These results also provide fundamental information for prudent harvesting strategies of wild donor coral colonies to generate stock for aquaculture, the aquarium trade (*Olivotto et al., 2011*; *Leal et al., 2016*), and translocation to degraded reef sites (*Miyazaki, Keshavmurthy & Funami, 2010*; *Toh et al., 2013*). Coral fragmentation and translocation corals is a highly stressful event, and the speed at which recovery occurs is critical to survival and ongoing health (*Lirman, 2000b*; *Roff, Hoegh-Guldberg & Fine, 2006*). Since storage lipids and their constituent FA represent important energy reserves during stressful periods for coral (*Imbs & Yakovleva, 2011*; *Denis et al., 2013*), judicious selection of branches inherently rich in these compounds will maximise their survival and recovery prospects. Indeed, it has been shown that there is a significant, inverse relationship between initial coral lipid stores and the onset timing of high mortality rates following major stress events (*Anthony et al., 2009*). Concordantly, branches undergoing rapid growth, which contain lower storage lipid reserves, have shown lowered regenerative capacities (*Oren et al., 2001*; *Roff, Hoegh-Guldberg & Fine, 2006*; *Denis et al., 2013*). Considering this, the results of the present study suggest that the L-66 position of the larger size class ($4,376 \pm 741$ cm$^2$), and the S-centre and S-50 positions of the smaller size class ($1,410 \pm 88$ cm$^2$), are metabolically-active branches containing large stores of energy-rich lipids and not undergoing rapid growth—thus representing positions with the greatest survival and health prospects following fragmentation and translocation.

# CONCLUSION

These results clearly show that the scleractinian coral colony is not nutritionally homogeneous. While different colony regions are functionally specialised for specific roles, so too are their nutritional profiles different, demonstrating tight integration of a single coral colony such that, while individual regions must undergo a trade-off for resource allocation, the colony as a whole is able to undergo several important life functions at once.

# ACKNOWLEDGEMENTS

The authors thank the SeaSim team at AIMS and the staff of Deakin University's School of Life and Environmental Sciences for technical assistance throughout the project. This work conforms to the legal requirements of Australia.

### Funding
The authors received no funding for this work.

### Competing Interests
The authors declare there are no competing interests.

### Author Contributions
- Jessica A. Conlan conceived and designed the experiments, performed the experiments, analyzed the data, wrote the paper, prepared figures and/or tables, reviewed drafts of the paper.
- Craig A. Humphrey and Andrea Severati conceived and designed the experiments, reviewed drafts of the paper.
- David S. Francis conceived and designed the experiments, wrote the paper, reviewed drafts of the paper.

### Field Study Permissions
The following information was supplied relating to field study approvals (i.e., approving body and any reference numbers):

Field collections were approved by the Great Barrier Reef Marine Park Authority: G12/35236.1.

### Data Availability
The raw data has been provided as Supplemental Information 1.

### Supplemental Information
Supplemental information for this article can be found online at http://dx.doi.org/10.7717/peerj.4239#supplemental-information.

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
