# Peer review of "Intra-colonial diversity in the scleractinian coral, Acropora millepora: identifying the nutritional gradients underlying physiological integration and compartmentalised functioning"

_PeerJ, doi:10.7717/peerj.4239_

## Round 0.1 · original submission · Major Revisions

· Academic Editor

Major Revisions

The reviewers found of high relevance your work indicating the potential contribution that this study has to improve coral restoration efforts. However, constructive suggestions have been given to improve the manuscript. One important point brought by one of the reviewers is that the authors need to focus on conclusion that are indeed supported by the data and analysis conducted. At the current stage it seems that some conclusion are not well supported by the results from statistical analyses.

In order to improve the flow and clarity of the paper, the authors require to make the discussion more focused and concise. The use of subheading would be beneficial.

Reviewer 1 ·

Basic reporting

See below

Experimental design

See below

Validity of the findings

See below

Additional comments

In this study, the authors examine nutritional gradients within coral colonies finding distinct heterogeneity in symbiont cell number, protein, lipid and energy content across the colony. The rationale for carrying out the study is strong. The authors state that this knowledge could contribute to coral nursery restoration practices where fragmenting colonies is key to propagation and increasing survival of fragments increases yield.
Overall the work is strong. The experimental design and methods are well done and described. The results are backed up with appropriate statistical rigor. I believe the study is a good contribution to coral physiology and restoration science.
I do feel, however, that the manuscript could be much stronger than it is currently with some work on the writing. To me, the message of this work is fairly straightforward and could be told in a much, much shorter manuscript. Succinct, but important, results call for a relatively short paper. I put this work into that category. The Introduction and especially the Discussion are overly long. The authors can tell a more succinct and better story. In the discussion, in every section, the authors recap what they have already said in the results. Get rid of these! You don’t need to say things twice. If you are struggling with this, then combine the results and discussion together. The discussion desperately needs subheadings. The discussion of the lipids is way too detailed and strays from the central theme of the paper. I believe that much of this could be omitted.
Another problem that makes the lipid section of the results remote to the average reader (me) is the lack of any background on lipid chemistry. The authors need to think about how to bring the uninitiated up to speed in short order before they dive into their data. I would suggest a simple table describing the lipids discussed, whether they are structural or storage etc. This could go in supplemental materials. Without this background it is very hard to follow this section.
Minor concerns/suggestions
1. Figure 1 is very important to understand how the sampling was done. But the authors could add a section somewhere – intro, or results and figure legend? – that describes in words how A. millepora grows – as multiple finger-like branches arrayed roughly in 2 dimensions. This will help the reader understand how the branched were sampled.
2. Put N numbers in the figures
3. Discuss the parts of figure 2 in order. If you keep the text the same, then you need to rearrange the figure.
4. What does proximate analysis mean? I am not familiar with this term
5. Stony corals are not soft corals – they are distantly related. There are multiple citations in the manuscript where soft coral findings are lumped with stony coral findings. These either need to be called out as different or removed. For example, whereas there is ample evidence that symbionts are transported around soft coral colonies, there are no data in hard corals to suggest this. Very different biology between the two. Compare apples to apples.

Reviewer 2 ·

Basic reporting

The manuscript’s main idea and general focus are good. It has been a long-standing goal in coral biology to understand the relationships between polyps within a colony. Specifically, It has been a major focus to understand if and when there are nutritional allocations within the colony that allow for subfunctionalization of different parts of the colony.

The authors do a good job summarizing the motivation of their experiments in the introduction. However, some of the statements asserted as fact in the introduction are not supported by the references provided. For example, the reference provided in line 75 refers to a mass-spec experiment trying to determine the proteins that accumulate in the skeleton. Unless I am mistaken, this paper does not speak to whether or not “protein fuels tissue growth, organogenesis, and calcification.” The same can be said about line 272 and 273. The authors should take more care in writing the introduction and discussion not do overstate conclusions or observations made by other authors.

Generally, the data presented here suggest some significant differences in the nutrition allocation across the coral. However, many of the differences are marginally significant. Given the variance in the data (all error bars in figures are SE) and the degree of multiple testing within the paper, it is hard to interpret the biological significance of the findings.

Experimental design

The experimental design attempted to be replicative both technically and biologically. The authors conducted well motivated experiments. However, they saw high variance in most of their measurement precluding robust statistical significance.

Major comments:

The authors aimed to determine if there is differential nutritional compartmentalization along across coral colonies of two different size classes. The authors deploy a sophisticated set of analytical approaches to address this question. However, it is not clear based on the paper how many of the stated conclusions are supported by significant statistical tests (Examples of this are discussed below). It is important for the reader to know, which results are significant in both the original ANOVA and post-hoc tests. Please report the both of these P-values for each statement in the text to help clarify this for the reader. For example, the end of Line 205.

I would like to see raw data points plotted over the SE plots for Figure 2 and 3.

In general, it is not clear how meaningful discussing the trends in the data are when the underlying differences are not significant (especially when contradicted by other corals in a different size class).

For example:

Line 196-198: "Within the size classes, there were no significant differences in total lipid (P>0.05) (mg g sample-1) (Fig. 2a), although both edge positions contained lower lipid compared to the next, immediate position."

Line 198- 200: "A similar trend was apparent in zooxanthellae densities for the larger
colonies (cells cm2 -1) (Fig. 2e), whereby the density at the edge was again lower than the next immediate position."

Line 206-208: "The caloric enthalpies showed that, for both size classes, the edge and next immediate position contained the highest energy contents, while the S-centre and L-centre and L-33 positions were low (Fig. 2f)."

There are many examples of this within the text, and they should be removed or modified to avoid confusing the reader into thinking these trends are significant.

Validity of the findings

As stated above, more effort should be made to ensure the reader can easily determine, which statements and claims came from significant statistical analysis. These limitations of the study should be addressed directly in the discussion.

Minor comments:

Line 67: 'reasonable to hypothesise that this' - spelling

Line 129: Elaborate on air-spraying method.

Line 135-137: Redundant with next paragraph

Line 142: Not clear how to crush something in a mortar and pestle under a french press. Can you please elaborate?

Line 153: Is four digits the appropriate number of significant figures for this measurement?

Line 212: "The proportion of storage lipid in the larger colonies followed a similar.." It is not clear what this proportion is similar to based on the text.

Figure 1: It would be helpful to reorder the figure panels to match the order in which they are discussed.

---

## Round 0.2 · Minor Revisions

· Academic Editor

Minor Revisions

The authors need to be cautious and avoid any claim of meaningful trend that is not supported statistically. Revise the discussion keeping this in mind.

Address all the remaining minor comments addressed by the reviewers.

Reviewer 1 ·

Basic reporting

All looks good.

Experimental design

Looks good.

Validity of the findings

Looks good.

Additional comments

The authors have addressed my concerns/suggestions. I would still ask that they simply define 'proximate' in the manuscript - they don't have to remove it, but just define it for the uninitiated.

Reviewer 2 ·

Basic reporting

No Comment

Experimental design

No Comment

Validity of the findings

The authors did a fine job indicating significances in the updated manuscript.

However, I am still worried about discussion "trends" in data as "biologically meaningful." In many cases, there is no support that the means are actually different (i.e. not statistically significant).

I suggest removing these discussions from the paper. because if these experiments were done again, these trends could go in the opposite direction. Illustrating this, many of these trends, are not supported across the different coral size classes.

Additional comments

No Comment

---

## Round 0.3 · accepted · Accept

· Academic Editor

Accept

The authors have addressed satisfactorily all the comments from the reviewers.